# Mixed Spices at Culinary Doses Have Prebiotic Effects in Healthy Adults: A Pilot Study

**DOI:** 10.3390/nu11061425

**Published:** 2019-06-25

**Authors:** Qing-Yi Lu, Anna M. Rasmussen, Jieping Yang, Ru-Po Lee, Jianjun Huang, Paul Shao, Catherine L. Carpenter, Irene Gilbuena, Gail Thames, Susanne M. Henning, David Heber, Zhaoping Li

**Affiliations:** Center for Human Nutrition, Department of Medicine, David Geffen School of Medicine, University of California at Los Angeles, Los Angeles, CA 90095, USA; ARasmussen@mednet.ucla.edu (A.M.R.); JiepingYang@mednet.ucla.edu (J.Y.); RuPoLee@mednet.ucla.edu (R.-P.L.); JianjunHuang@mednet.ucla.edu (J.H.); PShao@mednet.ucla.edu (P.S.); CCarpenter@mednet.ucla.edu (C.L.C.); IGilbuena@mednet.ucla.edu (I.G.); GThames@mednet.ucla.edu (G.T.); SHenning@mednet.ucla.edu (S.M.H.); dheber@mednet.ucla.edu (D.H.); zli@mednet.ucla.edu (Z.L.)

**Keywords:** spices, gut microbiota, prebiotic effect, firmicutes, short-chain fatty acids, dietary intervention

## Abstract

Spices were used as food preservatives prior to the advent of refrigeration, suggesting the possibility of effects on microbiota. Previous studies have shown prebiotic activities in animals and in vitro, but there has not been a demonstration of prebiotic or postbiotic effects at culinary doses in humans. In this randomized placebo-controlled study, we determined in twenty-nine healthy adults the effects on the gut microbiota of the consumption daily of capsules containing 5 g of mixed spices at culinary doses by comparison to a matched control group consuming a maltodextrin placebo capsule. The 16S ribosomal RNA sequencing data were used for microbial characterization. Spice consumption resulted in a significant reduction in Firmicutes abundance (*p* < 0.033) and a trend of enrichment in Bacteroidetes (*p* < 0.097) compared to placebo group. Twenty-six operational taxonomic units (OTUs) were different between the spice and placebo groups after intervention. Furthermore, there was a significant negative correlation between fecal short-chain fatty acid propionate concentration and Firmicutes abundance in spice intervention group (*p* < 0.04). The production of individual fecal short-chain fatty acid was not significantly changed by spice consumption in this study. Mixed spices consumption significantly modified gut microbiota, suggesting a prebiotic effect of spice consumption at culinary doses.

## 1. Introduction

An increasing body of evidence suggests that the gut microbiota has a profound impact on human health. While the microbiome of a healthy individual is relatively stable, gut microbial dynamics can be influenced by host lifestyle and dietary choices [1]. An acute change in dietary pattern from animal-based to plant-based diet alters gut microbial populations within 24 h and then reverts to baseline within 48 h of returning to the baseline dietary pattern [2]. Studies that involve intake of a specific dietary component demonstrate how certain microbiota tend to respond to nutrient-specific challenges. Protein, fats, digestible and non-digestible carbohydrates, probiotics, and dietary polyphenols all induce shifts in the microbiome with secondary effects on host immunological and metabolic markers [1,3]. An emerging and rapidly growing scientific literature is implicating the microbiome in a number of conditions and disorders including inflammatory bowel disease, obesity, type 2 diabetes mellitus, cardiovascular disease, cancer, autism, mood and neurodegenerative disorders [4,5,6,7,8]. 

The consumption of polyphenol-rich foods, including fruits and vegetables, has been reported to reduce pathogenic *Clostridia* and to enrich beneficial bacteria such as *Bifidobacterium* and *Lactobacillus* species in human studies [9,10]. In conjunction with these changes, reductions in plasma triglycerides and C-reactive protein have been noted [9]. Dietary polyphenols have been shown to help maintain intestinal health by preserving the gut microbial balance through the stimulation of the growth of beneficial bacteria and the inhibition of pathogenic bacteria, exerting prebiotic-like effects. 

We have previously demonstrated the effect of water extract of culinary spices, including cinnamon, Mediterranean oregano, ginger, rosemary, black and cayenne pepper, on the growth of 33 *Bifidobaterium* and *Latobacillus* spp., and its antimicrobial activity against 88 intestinal, pathogenic, and toxigenic bacterial strains in an in vitro model. These spices promoted the growth of *Bifidobaterium* and *Latobacillus* spp. Cinnamon, ginger, oregano, black pepper, and cayenne pepper showed activity against several pathogenic *Fusobaterium* and *Ruminococcus* spp. and selected *Clostridium difficile* strains [11]. The present pilot study was designed to investigate the effects of mixed spices (cinnamon, oregano, ginger, black pepper, and cayenne pepper) at culinary doses consumed over 2 weeks in a standardized 5 g capsule on the production of gut microbiota and short-chain fatty acids (SCFAs) in healthy subjects compared to a placebo maltodextrin capsule in a parallel randomized controlled clinical trial. 

## 2. Materials and Methods

### 2.1. Study Design and Spice Intervention

The pilot study was conducted in accordance with the guidelines of the Office of the Human Research Protection Programs (OHRPP) of the University of California, Los Angeles. The clinical protocol was approved by the UCLA Internal Review Board (IRB) and the study was registered at the NIH Clinical Trial Registry (NCT03676803). A total of 66 healthy women and men aged 18 to 65 were screened in 2017 through local advertisement. Participants with a history of gastrointestinal surgery, diabetes mellitus, or other serious medical conditions such as chronic hepatic or renal disease, bleeding disorder, congestive heart disease, chronic diarrhea, myocardial infarction, coronary artery bypass graft, angioplasty within 6 months prior to screening, current diagnosis of uncontrolled hypertension or chronic gastrointestinal disorders, bulimia, anorexia, laxative abuse, or endocrine disorders were excluded. Participants who were consuming a high-fiber/polyphenol diet; taking any medication or dietary supplement interfering with the absorption of polyphenols; pregnant or breastfeeding; frequently using prebiotics, probiotics, yogurt, or fiber supplements; taking antibiotics or laxatives within the previous 3 months; or currently using tobacco products were also excluded. Thirty-one subjects meeting enrollment criteria were recruited and provided written informed consent before the study began. A randomized, placebo-controlled, double-blind pilot study was conducted. Subjects were randomized according to an algorithm modified from a previous publication [12]. The study was divided into two periods: an initial run-in period of 1 week and an intervention period of 2 weeks. After the run-in period, 31 subjects were randomly allocated to consume either 5 g capsules of spice mixture containing 1 g (20%) cinnamon, 1.5 g (30%) oregano, 1.5 g (30%) ginger, 0.85 g (17%) black pepper, and 0.15 g (3%) cayenne pepper, or 5 g capsules containing maltodextrin daily for 2 weeks. The individual spices were purchased from local grocery stores.

At the start of study, participants received a diet instructional handout and were counseled by a registered dietitian on compliance. During the run-in and intervention period, all participants consumed a beige diet (low fiber < 10 g and low polyphenols < 3 servings of polyphenol rich fruit/vegetables per day). The beige diet handout advised participants to eat foods beige in color and rich in simple carbohydrates like white breads/bagels, crackers, granola bars, rice, macaroni/pasta, yogurt, dairy, poultry, cereal, and bananas and to avoid foods high in polyphenols and/or fiber. Participants were also provided with weekly checklists to track their fruit and vegetable intake throughout the study with a limit of three servings per day. This checklist included a list of higher fiber/polyphenol foods to avoid as well as a list of lower fiber/polyphenol fruits and vegetables they could consume as a part of their three servings per day, with a serving size estimation tool included. At the baseline and final visits, participants also completed and returned 3-day food records that were evaluated by the dietitian for compliance with the beige diet.

At baseline and at the end of 2 weeks of intervention period, body weight, body mass index (BMI) and composition were determined. Body composition was measured using the Tanita-BC418 bioelectrical impedance analyzer (Tanita Corp., Tokyo, Japan). Height was measured without shoes using a stadiometer (Detecto-Medic; Detecto-Scales; Brooklyn, NY, USA) and recorded to the nearest 0.1 cm. In the meantime, subjects reported on their overall well-being at baseline and at the end of two weeks by completing the SF-36 general wellness questionnaire.

### 2.2. Sample Collection

Fasting blood, 24 h urine, and stool samples were collected from each participant at baseline and at week 2 follow-up visit. Blood in EDTA was centrifuged 1500× *g* for 10 min at 10 °C and plasma stored at −80 °C until analysis. Stool collection utilized Stool Collection Kit (Thermo Fisher Scientific, Waltham, MA, USA), as per instructions, and specimen was stored on ice pack in a freezer box for up to 24 h before delivery on ice packs to UCLA. Subjects’ self-reported stool type numbers from The Bristol Stool Chart were recorded for assessment of stool consistency, and subjects also completed the UCLA Digestive Disease Center Symptom Questionnaire for the assessment of gastrointestinal symptoms (gas, bloating, diarrhea, etc.). At UCLA, the specimen was processed immediately and the aliquots stored at −20 °C.

### 2.3. Determination of Fecal SCFA and Urinary Rosmarinic Acid

Aliquot of stool samples was diluted, acidified, and filtered, and SCFAs (acetic, propionic, butyric, and valeric acid) were quantified by gas chromatography flame ionization detection, as previously described [13]. SCFA standard mix was purchased from Sigma-Aldrich (St Louis, MO, USA). Urinary free rosmarinic acid as a biochemical marker of adherence to spice supplementation was measured by high-performance liquid chromatography according to published methods [11,14].

### 2.4. DNA Extraction, 16S rRNA Sequencing, and Taxonomic Assignment

DNA from stool was extracted using the MoBio power soil DNA isolation kit (MoBio Laboratories, Carlsbad, CA, USA). The quality and quantity of the DNA was confirmed using a Nanodrop 1000 (Thermo Fisher Scientific. Sequencing was performed at MR DNA (www.mrdnalab.com, Shallowater, TX, USA) on a MiSeq (Illumina, San Diego, CA, USA), following the manufacturer’s guidelines. The minimum DNA concentration was 10 ng/μL. Negative controls were analyzed with the samples. The phiX ratio was a minimum of 20%. The 16S rRNA gene V4 variable region PCR primers 515/R806 with barcode on the forward primer were used in a 28 cycle PCR using the HotStarTaq Plus Master Mix Kit (Qiagen, Germantown, MD, USA) under the following conditions: 94 °C for 3 min, followed by 28 cycles of 94 °C for 30 s, 53 °C for 40 s, and 72 °C for 1 min, after which a final elongation step at 72 °C for 5 min was performed. After amplification, PCR products were checked in 2% agarose gel to determine the success of amplification and the relative intensity of bands. Sequence data were processed using MR DNA analysis pipeline (MR DNA). In brief, sequences were joined and depleted of barcodes, then sequences <150 bp and sequences with ambiguous base calls were removed. Sequences were then denoised, operational taxonomic units (OTUs) generated, and chimeras removed. OTUs were defined by clustering at 3% divergence (97% similarity). Final OTUs were taxonomically classified using BLASTn against a curated database derived from Greengenes 12_10 [15].

### 2.5. Identification of Changes in OTU Abundance Associated with Intervention 

To explore the changes of microbial composition during intervention, all analyses were conducted in R (version 3.5.2) (The R Foundation, Vienna, Austria) [16] with applied packages, including “phyloseq” [17], “ggplot2” [18], “vegan” [19], and “DESeq2” [20]. An OTU count table, taxonomy classification table, related clinical and demographic data, and the OTU phylogenetic tree were imported and analyzed as a “phyloseq” object. Measures of beta-diversity were computed using the weighted and unweighted UniFrac distance metric using phyloseq in R. Differences in whole communities across groups were determined by Permutational Multivariate Analysis of Variance (PERMANOVA) using the Adonis command provided by Vegan in R. The results were visualized via Principal Coordinate Analysis (PCoA) ordination (ggplot2).

OTUs differentially abundant between follow up and baseline visits in subjects receiving spice and placebo interventions were identified using DESeq2 [20]. This algorithm performs normalization using size factors estimated by the median-of-ratios method, employs an empirical Bayesian approach to shrink dispersion, and fits the data to negative binomial models [21]. Differential abundance was determined using the Wald test with automatic filtering of low abundance OTUs and automatic calculation of adjusted *p*-values (Bonferroni correction), and the enriched OTUs were visualized using the ggplot2 package in R. As Bonferroni correction is often considered overly conservative, an extended set of OTUs significant at *p* < 0.2 was listed.

Differentially abundant OTUs between spice and placebo groups were also detected by DESeq2. Null and test models were constructed for each OTU using DESeq. Each model includes variables “time” (baseline and follow up) and “intervention” (placebo or spice). Differentiating the two models was an interaction between “time” and “intervention”, which was only present in the test model. To identify OTUs with abundance changes that occurred specifically with spice but not placebo intervention, we used a likelihood ratio test (LRT) to test for a significant additional contribution of the interaction between time and spice vs. placebo exposure on OTU abundance in the test model when compared to the null model.

### 2.6. Statistical Analysis

Descriptive statistics, such as mean ± SEM, were used to summarize subjects’ demographic characteristics and biochemical results. Regression was calculated using LINEST function in Microsoft Excel 2010 (Microsoft Corporation, Redmond, WA, USA). Statistical difference of alpha diversity (Chao1 and Shannon), fecal SCFAs, and bacterial abundance at phylum levels were evaluated between interventions and overtime using a mixed model ANOVA, as there was a mixture of between groups (spice and placebo) and repeated measures (baseline and follow-up) variables using IBM SPSS Statistics version 23 (IBM, Armonk, NY, USA). Statistical significance was accepted at *p* ≤ 0.05.

## 3. Results

### 3.1. Subjects

Sixty-six healthy adults aged 18–65 years were screened, and 31 participants who met the enrollment criteria were recruited, randomized, and completed the 2-week study (Figure 1). Data of one subject from placebo group and one from spice group who did not comply with protocol was excluded based on the result of urinary rosmarinic acid analysis. Hence the mean concentration of rosmarinic acid in 24 h urine was 65.1 ± 63.8 μg from spice group (*n* = 14) and 0 ± 0 in placebo group (*n* = 15). Table 1 lists subjects’ baseline characteristics of placebo group and spice intervention group. Placebo group included 4 men and 11 women, had a mean BMI of 26.9 ± 4.5, had body fat of 30.0 ± 7.9%, and were aged 36.7 ± 13.3 years; spice intervention group included 4 men and 10 women, had a BMI of 28.2 ± 7.0, had body fat of 32.0 ± 12.5% and were aged 34.4 ± 12.5 years. In both groups, percent of body fat in males was lower than females. There was no statistical difference between groups for each variable.

### 3.2. Body Mass Index, Body Composition, Wellness Score, and Bristol Stool Scores

After a 2-week intervention period, the body weight and percent of body fat remained unchanged compared to baseline for both groups. There was no significant difference in general well-being scores as well as Bristol stool type and frequency. Two participants in spice group reported transient bloating, nausea, and stomach discomfort at the beginning of supplementation.

### 3.3. Microbial Composition

Alpha-diversity (i.e., diversity within a sample) for the spice and placebo group is shown in Figure 2. Supplementation of spice did not significantly affect α diversity indices of richness as measured by Chao 1 (Figure 2A) and richness and evenness as measured by Shannon index (Figure 2B). The unweighted and weighted UniFrac distance metric were calculated and visualized via principle coordinate analyses (PCoA) (Figure 3A,B). Neither weighted nor unweighted UniFrac distance metric showed a distinct separation between baseline and follow-up visits in placebo or spice groups (Figure 3A,B).

The changes in relative abundance at phylum level following the 2-week spice or placebo capsule consumption are illustrated in Figure 4. The phylum Firmicutes showed a 6% increase in mean abundance in placebo group and a 10% decrease in spice group (*p* = 0.033) over 2 weeks. Bacteroidetes abundance was elevated in spice group by 19% and reduced by 14% in the placebo group (*p* = 0.097). The mean abundance of actinobateria, verrucomicrobia, proteobacteria, fusobacteria, euryarchaeota, spirochaetes, tenericutes, cyanobacteria, and lentisphaerae was not significantly different between groups.

Twenty-six OTUs were significantly different in abundance between the groups (*p* < 0.02 using DESeq2 model) after 2 week supplementation. Among them, twenty-one OTUs displayed increased abundance and five showed reduced abundance indicated as fold change (Table 2). Changes in the abundance of sixteen OTUs of the phylum Firmicutes, four OTUs of the Bacteroidetes, four OTUs of the Proteobacteria, and two OTUs of the Actinobacteria were observed (*p* < 0.02). At family level, the most diverse change of OTUs are within family of *Lachnospiraceae* (*n* = 4) and *Ruminococcaceae* (*n* = 4), followed by *Streptococcaceae* (*n* = 3), *Clostridiaceae* (*n* = 2), *Erysipelotrichaceae* (*n* = 1), *Lactobacillaceae* (*n* = 1), and *Veillonellaceae* (*n* = 1). The increasing abundance in *Bacteroidaceae* (*n* = 4) was seen in all four *Bacteroides* species. The change in phylum Proteobacteria (*n* = 4) contained two genera and another two classified to the class level. There was significant enrichment of *Bifidobacterium animalis* (24-fold increase, *p* < 0.004), *Bacteroidetes fraglilis* (8-fold increase, *p* < 0.004) and *Lactobacillus* genus (79-fold increase, *p* < 0.002), and reduction in *Clostridium* genus (19-fold decrease, *p* < 0.005). Significance with adjusted *p*-values (*p* < 0.05 using Bonferroni correction) was found in *Bacteroides* (a 13-fold increase, OTU_583656) and in *Ruminococcus* (a 158-fold increase, OTU_359950).

### 3.4. Fecal Short Chain Fatty Acids and Correlation Analysis

Changes in each of main SCFA excretion from baseline to 2 weeks in two groups are presented in Figure 5. The percent changes of acetate, propionate, butyrate, valerate, and the total amount were 14.6%, 20.7%, 6.2%, 18.2%, and 14.4% for spice group and 5.4%, −1.6%, 28.1%, 27.6%, and 8.2% for placebo group, respectively. All changes in the SCFA content in both spice and placebo group were not statistically significant. We then evaluated correlation between each SCFA and the relative abundance of Firmicutes and Bacteroidetes. There was a significant negative correlation between propionate concentration and Firmicutes abundance (*R*= −0.391, *p* < 0.04) and a trend of positive correlation between propionate concentration and Bacteroidetes abundance (*R* = 0.435, *p* < 0.07) in spice intervention group (Table 3).

## 4. Discussion

Here, we report that daily intake of 5 g of mixed spices for 2 weeks in healthy subjects resulted in a significant reduction in the relative abundance of the phylum Firmicutes (*p* = 0.033), and a trend of increasing in phylum Bacteroidetes (*p* = 0.097) as compared with a matched control group. Most healthy adult microbiota are dominated by these two bacterial phyla, which together make up about 95% of gut microbiota [22]. A significantly higher abundance of Firmicutes and a higher Firmicutes/Bacteroidetes ratio has been frequently demonstrated in obese individuals [6,22,23], whose gut microbiome is characterized by an increased capacity for energy harvest, inflammation, and gut barrier disruption [24,25]. 

In addition to the shifts observed in Firmicutes and Bacteroidetes abundance, a number of OTUs in the family of *Bifidobateriaceae* and *Bacteroidaceae*, as well as the family of *Streptococcaceae*, *Lachnospiraceae*, *Ruminococcaceae*, *Veillonellaceae,* and *Erispelotrichaceae,* were significantly altered between groups (Table 2). Spice intervention significantly enhanced two *Bifidobacterium* OTUs of the Bifidobacteriaceae family—*B. animalis* and *B. pseudolongum*—as well as one *Lactobacillus* OTU compared with control group. *Bifidobacterium* has been shown to associate with the production of a number of potentially health promoting metabolites, including short chain fatty acids, conjugated linoleic acid, and bacteriocins [26]. The abundance of *B. animalis* was reported to negatively associate with the body mass index [27]. Results from the present study are consistent with published reports in that consuming polyphenol rich foods increases the relative abundance of *Bifidobacterium* and *Lactobacillus* and reduces pathogenic *Clostridium species* [9,28,29]. *Bifidobacterium* and *Labtobacillus* are considered “healthy bacteria”, and members of the *Bifidobacterium* and *Lactobacillus*, as well as *Streptococcus,* are frequently used as probiotic strains with evident health benefits such as immune-modulation, cancer prevention, inflammation management, and the control of diverse bacterial consortia infections [3,30,31].

We noted that the abundance of all three *Ruminococcus* spp. were increased. In humans, *Ruminococcus* spp. were found as abundant members of a “core gut microbiome” in a majority of humans [32]. Some *Ruminococcus* spp. in our gut microbiomes play an important role in helping us degrade and convert complex polysaccharides into a variety of nutrients for their hosts [33]. The slow digestion of these special carbohydrates has been associated with numerous health benefits such as reversing infectious diarrhea and reducing the risk of diabetes and colon cancer [34]. Interestingly, the *Bacteroides* spp., such as *B. fragilis*, also have the ability to recognize and metabolize plant- and host-derived polysaccharides, and to produce polysaccharides as well [35,36,37]. Polysaccharide A, e.g., synthesized by *B. fragilis,* can promote immunological tolerance to pathogenic species such as *Helicobacter hepaticus* and protect the host from inflammation and associated colorectal cancer [35,38].

SCFAs constitute approximately 10% of the energy source in healthy people. These microbial-derived products are utilized by the host and exert a range of health-promoting functions. Butyrate is used preferentially as an energy source by the gut mucosa, is anti-inflammatory, and protects against colorectal cancer [39], whereas propionate is largely taken up by the liver and is a good precursor for gluconeogenesis, and promotes satiety and reduction in cholesterol liponeogenesis and protein synthesis [40,41]. Acetate is the most predominant gut-produced SCFA in peripheral blood and plays a role in prevention of weight gain through an anorectic effect, inflammation, and metabolic dysregulation [42,43,44]. Valerate is present in substantially low amount, and research on this SCFA is very limited. A recent study, however, showed that valerate significantly inhibited the growth of *C. difficile* in vitro and in vivo, suggesting valerate can potentially be used as a safe, microorganism-free method to treat *C. difficile* infection [45]. In the present work, the level of all four SCFAs was trending higher after spice intervention but the difference did not attain statistical significance due perhaps to the large interpersonal variations and small size of the study population. Nonetheless, our results indicated that spice supplementation can change SCFA production. 

Human studies investigating the effects of spice supplementation on gut microbiota composition are very limited. Kang et al., reported that intake of capsaicin powder in healthy subjects increased the Firmicutes/Bacteroidetes ratio, accompanied with increased plasma levels of glucagon-like peptide 1 and gastric inhibitory polypeptide and decreased plasma ghrelin level [46]. In animal models, Van Hul et al., reported that cinnamon bark extract lowered fat mass gain and adipose tissue inflammation in mice fed a high-fat diet leading to reduced liver steatosis and lower plasma nonesterified fatty acid levels that were associated with change on the microbial composition [47]. Spices have been shown to alter serum biochemical parameters related to inflammation or low-grade inflammation induced by high-fat diet [47,48,49] and have protective effects against chronic diseases [50,51]. This evidence indicates that spices may play important role in modulating the growth of gut microbiota and promoting human health.

The observed lack of significant difference in microbial richness is largely attributed to the brevity of the intervention, which is also believed to account to for the lack of change in species evenness as well as of overall microbial composition. Another limitation of the study is that to avoid the sustained effects [52] of spices on microbiota, our study design was a randomized placebo-controlled and not a crossover clinical trial, therefore, there are large interpersonal variations in gut microbiota composition and metabolite (SCFA) formation. Lastly, since this is not a controlled feeding study, dietary recall data analyzed by a dietitian was the only method used to assess participants’ compliance with the beige diet. Analysis of dietary data may have an impact on interpretation of the results.

## 5. Conclusions

The major significance of the current pilot study was that it revealed in humans for the first time that a mixture of spices at culinary doses affects the composition of gut microbiota. These observations also suggest but do not yet demonstrate an effect of spices on the metabolic activity of gut bacteria. The inverse relationship between fecal propionate and the phylum Firmicutes suggests that spices may be affecting the production of short-chain fatty acids, but our observations did not find statistically significant changes in SCFA levels. Nonetheless, the health benefits attributed to spices throughout ancient times may be proven in future studies of the prebiotic effects of spices in humans on glucose metabolism, inflammation, and cognition.

## Figures and Tables

**Figure 1 nutrients-11-01425-f001:**
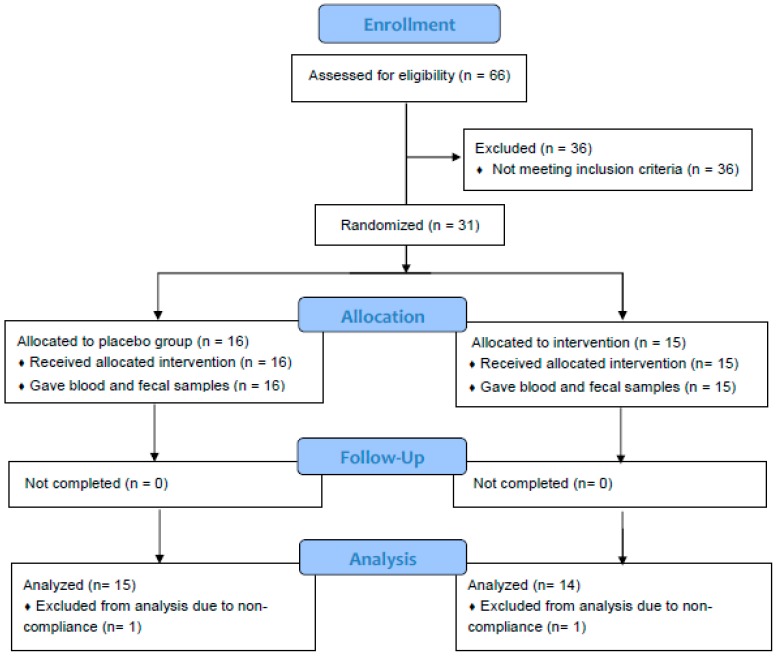
Flowchart of study subjects. A total of 31 subjects were randomized and completed study. Twenty-nine subjects were included in data analysis. Two subjects were excluded due to non-adherence to protocol.

**Figure 2 nutrients-11-01425-f002:**
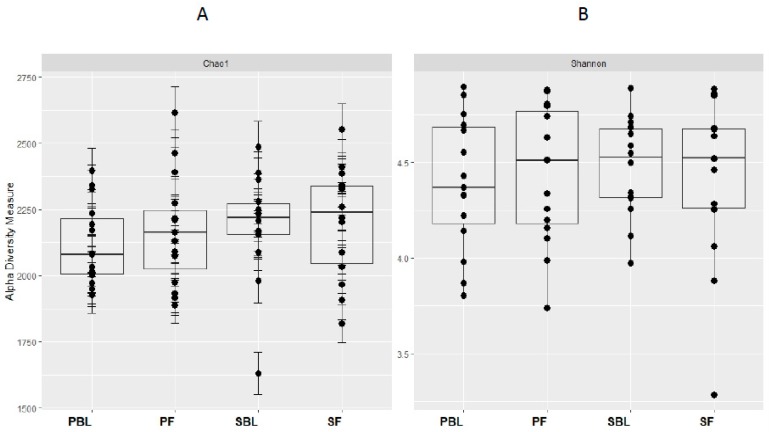
Within-sample alpha-diversity of fecal of 29 subjects collected at the baseline and 2 weeks. Calculation of the alpha-diversity for each sample for evaluating species richness and diversity by using Chao1 (**A**) and Shannon (**B**) effective indices. The diversity of a microbial profile for a certain index is the number of different species related to abundance and richness. PBL, placebo group at baseline; PF, placebo group at follow up; SBL, spice group at baseline; and SF, spice group at follow up.

**Figure 3 nutrients-11-01425-f003:**
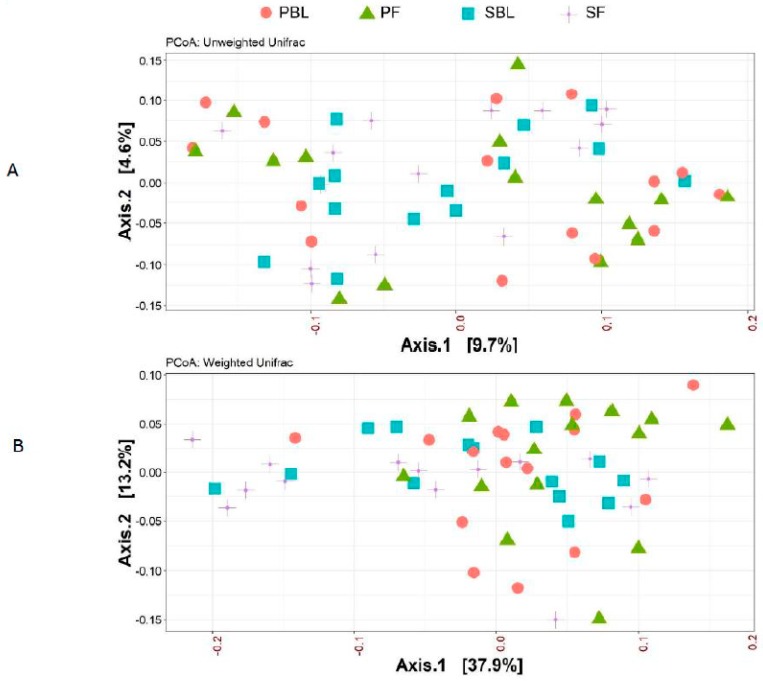
Beta-diversity between placebo group and spice group. Principal coordinate analysis (PCoA) plots of all fecal samples from 29 subjects in two groups at baseline and 2 week follow up. Red circle and green triangle represent participants in PBL and PF, blue square and purple cross represent participants in mixed SBL and SF, respectively. (**A**) Unweighted UniFrac. (**B**) Weighted UniFrac. Axis 1, principal coordinate 1; axis 2, principal coordinate 2. PBL, placebo group at baseline; PF, placebo group at follow up; SBL, spice group at baseline; and SF, spice group at follow up.

**Figure 4 nutrients-11-01425-f004:**
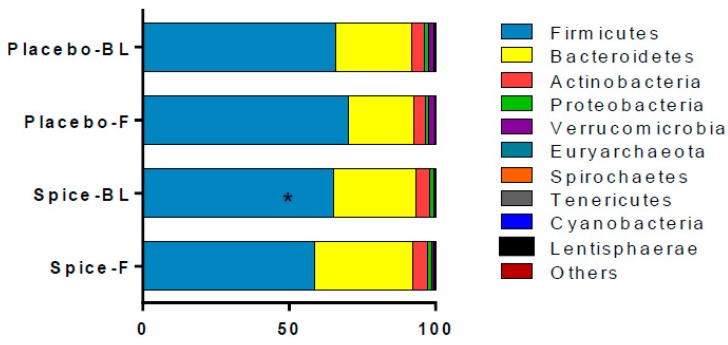
Comparison of fecal microbiota in subjects supplemented with placebo and mixed spices capsules. Bar graph shows the relative abundance of phylum in subjects supplemented with placebo (*n* = 15) or mixed spices (*n* = 14) at baseline and after 2 week intervention. Each color within the bar graphs represents a phylum, with area of the color proportional to relative abundance. PBL, placebo group at baseline; PF, placebo group at follow up; SBL, spice group at baseline; SF, spice group at follow up. Data are means ± standard errors (SE). Significant difference in abundance between placebo and spice group is indicated by * *p* < 0.05.

**Figure 5 nutrients-11-01425-f005:**
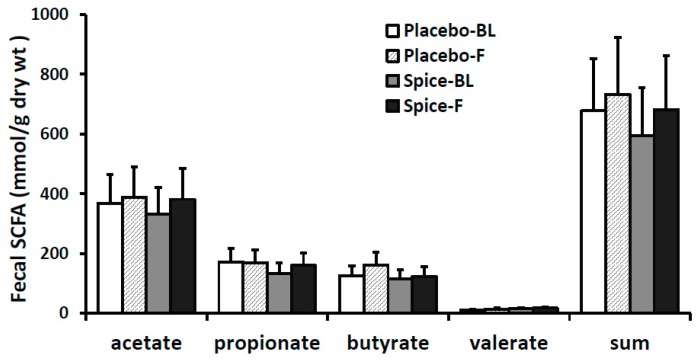
Fecal short-chain fatty acid production in subjects supplemented with placebo (*n* = 15) or mixed spices (*n* = 14) at baseline and after 2 week intervention. Data are means ± standard errors (SE). BL, baseline; F, follow up.

**Table 1 nutrients-11-01425-t001:** Selected baseline characteristics of study participants.

	Placebo Group	Spice Group	*p*-Value
Enrolled/randomized	15	14	
Age, years	36.7 ± 13.3	34.4 ± 12.5	0.63 ^a^
Sex			
Male (*N*)	4	4	
Female (*N*)	11	10	0.91 ^b^
Race			
African American (*N*)	1	3	
White (*N*)	9	6	
Asian (*N*)	3	3	
Multi-Racial (*N*)	2	2	
Other	0	1	0.57
Weight, kg	75.6 ± 16.9	76.7 ± 16.4	0.95
BMI, kg/m^2^	26.9 ± 4.5	28.2 ± 7.0	0.54
Body fat ^c^ (%)	30.0 ± 7.9	32.0 ± 12.5	0.63

Values are presented as mean ± SD. ^a^ For continuous data, *p*-value derived from *t*-tests. ^b^ For categorical data, *p*-value derived from Chi Square distribution tests. ^c^ Body fat (%) was determined by Tanita-BC418 bioelectrical impedance analyzer. Body mass index (BMI) was calculated by weight/height^2^.

**Table 2 nutrients-11-01425-t002:** Log_2_ fold changes (FC) for operational taxonomic units (OTUs) with differential abundance between the spice and placebo groups from baseline to 2 weeks.

Phylum/OTU No.	Class, Order, Family, Genus, and Species	log_2_ FC	*p*	*p* _adj_
**Actinobacteria**				
4426298	Actinobacteria; Bifidobacteriales; Bifidobacteriaceae; Bifidobacterium; animalis	4.578	0.004	0.496
681370	Actinobacteria; Bifidobacteriales; Bifidobacteriaceae; Bifidobacterium; pseudolongum	1.890	0.002	0.442
**Bacteroidetes**				
583656	Bacteroidia; Bacteroidales; Bacteroidaceae; Bacteroides;	3.675	0.000	0.037
175535	Bacteroidia; Bacteroidales; Bacteroidaceae; Bacteroides;	3.138	0.014	0.839
344525	Bacteroidia; Bacteroidales; Bacteroidaceae; Bacteroides; eggerthii	3.045	0.013	0.839
351231	Bacteroidia; Bacteroidales; Bacteroidaceae; Bacteroides; fragilis	2.949	0.004	0.498
**Firmicutes**				
679245	Bacilli; Lactobacillales; Lactobacillaceae; Lactobacillus;	6.297	0.002	0.442
1033473	Bacilli; Lactobacillales; Streptococcaceae; Streptococcus;	3.138	0.002	0.442
579608	Bacilli; Lactobacillales; Streptococcaceae; Streptococcus;	2.899	0.002	0.442
828043	Bacilli; Lactobacillales; Streptococcaceae; Streptococcus;	2.865	0.003	0.496
780650	Clostridia; Clostridiales; Clostridiaceae;	2.175	0.021	0.988
558420	Clostridia; Clostridiales; Clostridiaceae; Clostridium;	−4.226	0.005	0.530
323778	Clostridia; Clostridiales; Lachnospiraceae;	−3.353	0.014	0.839
176450	Clostridia; Clostridiales; Lachnospiraceae; Blautia;	2.720	0.007	0.655
589313	Clostridia; Clostridiales; Lachnospiraceae; Blautia; producta	−4.178	0.005	0.568
297182	Clostridia; Clostridiales; Lachnospiraceae; Dorea;	3.264	0.020	0.988
647215	Clostridia; Clostridiales; Ruminococcaceae; Oscillospira;	−3.902	0.012	0.839
359950	Clostridia; Clostridiales; Ruminococcaceae; Ruminococcus;	7.302	0.000	0.002
193336	Clostridia; Clostridiales; Ruminococcaceae; Ruminococcus;	3.631	0.009	0.777
2979308	Clostridia; Clostridiales; Ruminococcaceae; Ruminococcus;	2.520	0.009	0.777
173744	Clostridia; Clostridiales; Veillonellaceae; Megasphaera;	2.129	0.014	0.839
587530	Erysipelotrichi; Erysipelotrichales; Erysipelotrichaceae; Eubacterium; dolichum	2.819	0.015	0.839
**Proteobacteria**				
4435655	Alphaproteobacteria;RF32;	3.483	0.010	0.778
211720	Alphaproteobacteria;RF32;	6.491	0.016	0.839
636296	Betaproteobacteria; Burkholderiales; Alcaligenaceae; Sutterella;	−3.958	0.016	0.839
865469	Gammaproteobacteria; Pasteurellales; Pasteurellaceae; Haemophilus; parainfluenzae	4.238	0.021	0.988

*p*, *p* value for using DESeq set at *p* < 0.02; *p*_adj_, adjusted *p*-values using Bonferroni correction.

**Table 3 nutrients-11-01425-t003:** Correlation between percent relative abundance of bacteria and SCFA concentration after 2 week spice intervention.

	Acetate		Propionate		Butyrate		Sum	
	*R*	*p*	*R*	*p*	*R*	*p*	*R*	*p*
Firmicutes	−0.254	0.27	−0.391	0.04	−0.210	0.28	−0.290	0.13
Bacteroidetes	0.287	0.14	0.435	0.07	0.216	0.27	−0.083	0.27

Abbreviation: regression (*R*); *p*, *p* value for statistical significance. Concentration of SCFA, μmol/g dry weight; Sum, total concentration of acetate, propionate, butyrate, and valerate.

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
