# Peer review of "Mixed Spices at Culinary Doses Have Prebiotic Effects in Healthy Adults: A Pilot Study"

_nutrients, 2019, doi:10.3390/nu11061425_

Reviewer 1 Report

Authors Lu et al. have investigate the effects of mixed spices (cinnamon, oregano, ginger, black pepper and cayenne pepper) at culinary doses consumed over 2 weeks in a standardized 5 gram capsule compared to a placebo maltodextrin capsule on gut microbiota and short-chain fatty acids (SCFAs) production in healthy subjects in a parallel randomized controlled  clinical trial. Overall the study is well conducted, however, authors needs to address few minor comments before publication.

Specifc comments:

Overall there is no mention of prebiotics? No correlation. Please add.

Authors have used culinary spice blend in various amounts such as 5 gram 83 capsules of spice mixture containing 1g (20%) cinnamon, 1.5g ( 30%) oregano, 1.5g (30%) ginger, 0.85g 84 (17%) black pepper and 0.15g (3%) cayenne pepper. Each spice was divided individually with different dose %. Why? Whether this is recommended human dose?

Table is not properly formatted or presented. 

Values are presented as mean "±" SD is missing. 

P-values are not aligned with the variables in the tables and leads tor misinterpretation. 

Regarding the body fat%, as the participants are mixed sex (male and female), an overall body fat% can be invalid. As body fat% of male and females differ. Hence needs to be corrected. 

The color combination of groups in PCoA plot is not clear. Needs to be changes. 

SCFA count is completely insignificant and feels normal. However, microbial changes occured. Can you please explain. ?

Author Response

Reviewer 1

Comments and Suggestions for Authors

Authors Lu et al. have investigate the effects of mixed spices (cinnamon, oregano, ginger, black pepper and cayenne pepper) at culinary doses consumed over 2 weeks in a standardized 5 gram capsule compared to a placebo maltodextrin capsule on gut microbiota and short-chain fatty acids (SCFAs) production in healthy subjects in a parallel randomized controlled  clinical trial. Overall the study is well conducted, however, authors needs to address few minor comments before publication.

Response to Reviewer 1:

We greatly appreciate reviewer’s comments and the thoughtful reviews, and the opportunity to revise our manuscript. We have addressed the reviewers’ concerns as detailed below:

Specific comments:

1. Overall there is no mention of prebiotics? No correlation. Please add.

Response:  This intervention study was a continuation of our previous in vitro study demonstrated the prebiotic effect of individual spice (cited as ref #11, titled “Prebiotic Potential and Chemical Composition of Seven Culinary Spice Extracts”).  Since there has not been a demonstration of prebiotic effects at culinary doses in humans (abstract), we conducted this research to investigate if our in vitro findings would translate into humans. Indeed, we found spice mix supplementation decreased Firmicutes abundance and increased beneficial bacteria and shifted of short chain fatty acid production (a postbiotic effect). Nonetheless, more spice intervention studies with larger sample size are needed to demonstrate health benefits.  Therefore, we concluded in the concluding remarks as: “the health benefits attributed to spice throughout ancient times may be proven in future studies of the prebiotic effects of spices in humans on glucose metabolism, inflammation, and cognition”.

 2.

Authors have used culinary spice blend in various amounts such as 5 gram capsules of spice mixture containing 1g (20%) cinnamon, 1.5g ( 30%) oregano, 1.5g (30%) ginger, 0.85g 84 (17%) black pepper and 0.15g (3%) cayenne pepper. Each spice was divided individually with different dose %. Why? Whether this is recommended human dose?

Response: Our goal is to study the most commonly consumed spices. We selected five spices out of seven based on their prebiotic activity in an in vitro study (see ref #11). We also considered the popularity of these spices to compose the percentage in the blend.

3.

Table is not properly formatted or presented. 

Values are presented as mean "±" SD is missing. 

P-values are not aligned with the variables in the tables and leads tor misinterpretation. 

Regarding the body fat%, as the participants are mixed sex (male and female), an overall body fat% can be invalid. As body fat% of male and females differ. Hence needs to be corrected. 

Response: We double checked the table, all signs and the alignment are correct. It is possible that the production process changed the format and presentation.   

Regarding the body fat % of male and female separately, reviewer made a good point in that % body fat may differ in male and female. In both control and spice group average body fat % of male is lower compared to that of female, which is common.  Since the study sample size is small and male sample size much small (n= 4 in both groups) we choose to add  information in the Results (under 3.1. Subjects) to include body fat % and to indicate average male body fat % is lower in both groups compared to female.

4.

The color combination of groups in PCoA plot is not clear. Needs to be changes. 

Response: We have replotted PCoA and used different geometric shapes to represent individuals in 4 groups. We are able now to distinguish them even in non-colored print out.

5.

SCFA count is completely insignificant and feels normal. However, microbial changes occured. Can you please explain?

Response:  There is emerging evidence that diet-driven changes in microbiota diversity lead to changes in SCFAs. In a recent diet-switch study, where African Americans were fed a high-fiber, low-fat African-style diet and rural Africans a high fat, low-fiber western-style diet, the investigators observed profound shifts in gut microbiota composition, and SCFA and bile acids in the fecal water (Holmes E et al. Gut microbiota composition and activity in relation to host metabolic phenotype and disease risk. Cell Metab 2012; 16:559-64). In addition to the dietary factor, the production of SCFA in humans is regulated by a number of host, environmental and microbial factors (Macfarlane S. et al. Regulation of short-chain fatty acid production. Proc Nutr Soc 2003; 62: 67–72).  The main factors that control SCFA production are thought to be interrelated and include type and amount of available substrate, composition of the gut microbiota and gut transit time. Previous work in healthy participants has also shown that fecal SCFA concentrations may reflect SCFA absorption rather than its production (Vogt JA. Wolever TMS. Fecal acetate is inversely related to acetate absorption from the human rectum and distal colon. J Nutr 2003; 133: 3145–3148). Therefore, fecal SCFA concentration is determined by not only microbial diversity but also production and absorption. Hence, the link between microbiome composition and SCFA production is rather more difficult to explain. Based on the available information, we can only speculate that spice intervention changed microbiota composition and altered colonic fermentation pattern leading to the increase in SCFA production (insignificanly).

Reviewer 2 Report

Dear Authors,

The Editor of Nutrients has kindly asked me to review your manuscript.  I hope my feedback will enable you to improve an already great paper. Congratulations. Please find my comments below.

Lines 42-45. 

The microbiome is also implicated in neurodegeneration / neuroprotection in humans. A simple PubMed search on “neuroprotective and microbiome” retrieves some good examples of papers that would merit citing to substantiate the addition of neurodegenerative disorders to this list”. Additionally, the term “cognition” is used in the conclusions section but isn’t mentioned anywhere else in the manuscript. Please see my comment on this under “Conclusions” below.

https://www.ncbi.nlm.nih.gov/pubmed/?term=neuroprotective+and+microbiome

Therefore, I suggest this paragraph be changed as follows:

An emerging and rapidly growing scientific literature is implicating the microbiome in a number of conditions and disorders including inflammatory bowel disease, obesity, type 2 diabetes mellitus, cardiovascular disease, cancer, autism, mood and neurodegenerative disorders.

Section 4. Discussion

Page 2 of 18 (apologies for the formatting, no lines were provided in the manuscript)

SCFAs constitute approximately 10% of the energy source in healthy people. These microbial-derived products are utilized by the host and exert a range of health-promoting functions. Butyrate is used preferentially as an energy source by enterocytes in the gut mucosa. It is anti-inflammatory and protects against colorectal cancer (reference needed), whereas propionate is largely taken up by the liver and is a good precursor for gluconeogenesis, promotes satiety and reduction in cholesterol liponeogenesis and protein synthesis (reference needed).[

Comment: No references have been provided for butyrate and proprionate, but some are provided for acetate. Citing some good quality clinical studies to back up facts about butyrate and proprionate will make this section more coherent.

Acetate is the most predominant gut-produced SCFA in peripheral blood and plays a role in prevention of weight gain through an anorectic effect, inflammation, metabolic dysregulation (Morison and Preston, 2016) [39, 40].

Comment: The Morison and Preston citation hasn’t been processed in MDPI style. Additionally, your SCFA discussion and analysis includes valerate, yet there’s no explanation as to what the function of this SCFA. The reader would benefit from a brief account on valerate's value as part of your analysis, along the lines of what you've done with the other SCFAs. 

Page 3 of 18 (no lines provided)

Due to the brief intervention period we were unable to detect significant difference in microbial richness and species evenness as well as overall microbial composition.

Comment: A thorough review of the literature in this subject would result in changing this statement to acknowledge the brevity of the intervention as a major limitation of the study.

I would strongly suggest to change this paragraph to reflect the above, e.g.:

The observed lack of significant difference in microbial richness is largely attributed to the brevity of the intervention, which is also believed to account to for the lack of change in species evenness as well as of overall microbial composition.

Another limitation of the study is that to avoid the sustained effects of spices on microbiota, our study design was a randomized placebo-controlled and not a crossover clinical trial, therefore, there are large interpersonal variations in gut microbiota composition and metabolite (SCFA) formation.

Comment: Is there any evidence of sustained effects of herbs and spices on the human microbiome or indeed in animal models? If that is the case, it would be appropriate to refer to these studies here. Otherwise this could be seen as an oxymoron.

Lastly, since this is not a controlled feeding study, dietary recall data were used only by a dietitian to assess participants’ compliance with beige diet. Analysis of dietary data may have an impact on interpretation of the results.

Comment: There is no other mention of dietary recall data in the manuscript. This data is all important in this type of intervention and is missing from the methods. How long was it collected for? What tool was used to collect it, i.e. FFQ (food frequency questionnaire) or similar? Was it a validated tool?

Author Response

Response to Reviewer 2

We greatly appreciate reviewer’s helpful comments and thoughtful reviews, and the opportunity to revise our manuscript. We have addressed the reviewers’ concerns as detailed below (I have added number for clarity):

1. Lines 42-45. 

The microbiome is also implicated in neurodegeneration / neuroprotection in humans. A simple PubMed search on “neuroprotective and microbiome” retrieves some good examples of papers that would merit citing to substantiate the addition of neurodegenerative disorders to this list”. Additionally, the term “cognition” is used in the conclusions section but isn’t mentioned anywhere else in the manuscript. Please see my comment on this under “Conclusions” below. 

https://www.ncbi.nlm.nih.gov/pubmed/?term=neuroprotective+and+microbiome

Therefore, I suggest this paragraph be changed as follows:  

An emerging and rapidly growing scientific literature is implicating the microbiome in a number of conditions and disorders including inflammatory bowel disease, obesity, type 2 diabetes mellitus, cardiovascular disease, cancer, autism, mood and neurodegenerative disorders.

Response: we completely agree with reviewer's suggestion and have revised the sentence by adding “mood and neurodegenerative disorders” according to the suggestion.

Section 4. Discussion 

2. Page 2 of 18 (apologies for the formatting, no lines were provided in the manuscript)

 SCFAs constitute approximately 10% of the energy source in healthy people. These microbial-derived products are utilized by the host and exert a range of health-promoting functions. Butyrate is used preferentially as an energy source by enterocytes in the gut mucosa. It is anti-inflammatory and protects against colorectal cancer (reference needed), whereas propionate is largely taken up by the liver and is a good precursor for gluconeogenesis, promotes satiety and reduction in cholesterol liponeogenesis and protein synthesis (reference needed).[

Comment: No references have been provided for butyrate and proprionate, but some are provided for acetate. Citing some good quality clinical studies to back up facts about butyrate and proprionate will make this section more coherent.

Response: we have added two references, one for butyrate (1) and one for propionate (2-3):

1. McNabney SM et al. Nutrients. (2017) Short Chain Fatty Acids in the Colon and Peripheral Tissues: A Focus on Butyrate, Colon Cancer, Obesity and Insulin Resistance. Nutrients.

2. Louis P, Flint HJ. Formation of propionate and butyrate by the human colonic microbiota. Environ Microbiol. 2017 Jan;19(1):29-41.

3. Canfora EE, Jocken JW, Blaak EE. Short-chain fatty acids in control of body weight and insulin sensitivity. Nat Rev Endocrinol. 2015 Oct;11(10):577-91

3. Acetate is the most predominant gut-produced SCFA in peripheral blood and plays a role in prevention of weight gain through an anorectic effect, inflammation, metabolic dysregulation (Morison and Preston, 2016) [39, 40].

Comment: The Morison and Preston citation hasn’t been processed in MDPI style. Additionally, your SCFA discussion and analysis includes valerate, yet there’s no explanation as to what the function of this SCFA. The reader would benefit from a brief account on valerate's value as part of your analysis, along the lines of what you've done with the other SCFAs. 

Response: Valerate is present in substantially low amount and research on this SCFA is very limited. A recent study, however, showed that valerate significantly inhibited the growth of C. difficile in vitro and in vivo, suggesting valerate can potentially be used as a safe, microorganism-free method to treat C. difficile infection (Julie A. K. McDonald, 2018). We have added the information and the reference by Morrison in the revised manuscript.

Page 3 of 18 (no lines provided)

4. Due to the brief intervention period we were unable to detect significant difference in microbial richness and species evenness as well as overall microbial composition. 

 Comment: A thorough review of the literature in this subject would result in changing this statement to acknowledge the brevity of the intervention as a major limitation of the study.

    I would strongly suggest to change this paragraph to reflect the above, e.g.:

The observed lack of significant difference in microbial richness is largely attributed to the brevity of the intervention, which is also believed to account to for the lack of change in species evenness as well as of overall microbial composition. 

Response:  we completely agree with the above comments and have revised the paragraph:

“The observed lack of significant difference in microbial richness is largely attributed to the brevity of the intervention, which is also believed to account to for the lack of change in species evenness as well as of overall microbial composition. Another limitation of the study is that to avoid the sustained effects (see response to #5) of spices on microbiota, our study design was a randomized placebo-controlled and not a crossover clinical trial, therefore, there are large interpersonal variations in gut microbiota composition and metabolite (SCFA) formation.  Lastly, since this is not a controlled feeding study, dietary recall data were used only by a dietitian to assess participants’ compliance with beige diet. Analysis of dietary data may have an impact on interpretation of the results.

5. Another limitation of the study is that to avoid the sustained effects of spices on microbiota, our study design was a randomized placebo-controlled and not a crossover clinical trial, therefore, there are large interpersonal variations in gut microbiota composition and metabolite (SCFA) formation.

Comment: Is there any evidence of sustained effects of herbs and spices on the human microbiome or indeed in animal models? If that is the case, it would be appropriate to refer to these studies here. Otherwise this could be seen as an oxymoron.

Response:   In an earlier study (1) by some co-authors in collaboration with Dr. Finegold, prebiotics xylooligosaccharide, or XOS, were consumed at low and high dose. The study consisted of a 2 week run in, an 8 week intervention, and a 2 week washout phase. Results were reported at baseline, 4 week, 8 week and 10 week. It was clear that after a two-week wash out period at 10 week, bacterial counts of Bifdobacterium, Anaerobes-total, Aerobes-total Bacteroides fragilis (Table 2) are all comparable with those at week 8, and significantly or borderline significantly higher compared to those of baseline data.

1. Finegold SM, Li Z, Summanen PH, Downes J, Thames G, Corbett K, Dowd S, Krak M, Heber D. Food Funct. 2014 Mar;5(3):436-45.

6. Lastly, since this is not a controlled feeding study, dietary recall data were used only by a dietitian to assess participants’ compliance with beige diet. Analysis of dietary data may have an impact on interpretation of the results.

Comment: There is no other mention of dietary recall data in the manuscript. This data is all important in this type of intervention and is missing from the methods. How long was it collected for? What tool was used to collect it, i.e. FFQ (food frequency questionnaire) or similar? Was it a validated tool?

Response: We used 3-day food record (original manuscript line 97-98, under the subtitle of 2.1) that was evaluated by our dietitian for the compliance with the beige diet.